# Nitrate Quantification in Fresh Vegetables in Shanghai: Its Dietary Risks and Preventive Measures

**DOI:** 10.3390/ijerph192114487

**Published:** 2022-11-04

**Authors:** Fan Luo, Xiao-Juan Yan, Xue-Feng Hu, Li-Jun Yan, Ming-Yang Cao, Wei-Jie Zhang

**Affiliations:** School of Environmental & Chemical Engineering, Shanghai University, Shanghai 200444, China

**Keywords:** vegetables, nitrate, nitrite, chemical fertilizers, storage

## Abstract

To investigate nitrate and nitrite content in fresh vegetables, 264 samples were randomly collected in the farmers’ markets in Shanghai, Southeast China. The results indicate that 25.0% of the fresh vegetables were critically or more contaminated by nitrate [>1440 mg/kg FW (Fresh weight)]. Generally, leafy vegetables were more highly enriched in nitrate than root-tuber and fruit vegetables. About 22.6% of the leafy vegetables had a nitrate content exceeding the limit for edible permission (>3000 mg/kg FW). Nitrite content in the fresh vegetables was all within the safe level (<1 mg/kg FW). It was estimated that the daily nitrate intake through eating vegetables in Shanghai exceeded the WHO/FAO allowable limit. The field experiment indicated that the hyper-accumulation of nitrate and nitrite in the vegetables was mainly attributed to the excessive application of chemical fertilizers. The maxima of nitrate and nitrite in the vegetables were attained one week after applying chemical fertilizer, and thus they cannot be picked for dietary use. Applying organic manure can effectively lower the risk of nitrate and nitrite contamination in vegetables. The old leaves and leaf petioles were more easily enriched in nitrate due to their weaker metabolic activity. Vegetables with high nitrate content had a high risk of nitrite toxicity during storage due to the biological conversion of nitrate into nitrite, which is easily triggered by suitable temperature and mechanical damage processing. Therefore, fresh vegetables should be stored by rapid cooling and in undamaged forms to prevent nitrite accumulation.

## 1. Introduction

Nitrate is often accumulated in vegetables due to excessive chemical fertilizers and unreasonable farming practices [1,2]. Sometimes, nitrate content in fresh vegetables can attain as high as several thousand mg/kg FW. About 80~95% of daily intakes of nitrates in the human body are derived from vegetable consumption [3].

In 2012, the amount of chemical fertilizer application in China was 5.84 × 10^10^ kg, making up 33% of the global total and three times the world average value [4]. However, the coefficient of chemical nitrogen utilization in China is only about 30%, of which the rest mostly enters the environment through runoff or leaching [5]. Farmers often try to gain as high a yield of vegetables as possible to raise profits by excessively applying chemical fertilizers, which harms the environment and causes hyper-accumulation of nitrate in vegetables [3,6]. The mechanism of nitrate accumulation in vegetables is still not clearly understood. Nitrate content in vegetables is not only controlled by external factors like fertilizer application, climate, plant diseases, and damage by pests; but is also highly influenced by internal factors such as vegetable varieties, genotypes, and vegetative tissues [7,8].

Nitrate intake from vegetables is bioavailable in the human body [9] and can be transformed into nitrite [10]. About 5~8% of nitrate from diet can be reduced to nitrite by microflora in the oral cavity [11]. Saliva contributes greatly to the daily intake of nitrite [12]. Therefore, since the 1960s, many previous studies have focused on nitrate accumulation in vegetables.

Excessive dietary intakes of nitrite and nitrate may lead to the diseases of gastriccarcinomaor methaemoglobinaemia [13]. Although the results are still controversial [14], it is indisputable that nitrite can be combined with dietary amines to form potentially carcinogenic N-nitroso compounds [15]. Nitrite also has direct physiological toxic effects on the human body [16]. A series of regulations, therefore, have been specified to limit the concentrations of nitrate and nitrite in vegetables and food [14].

The yield of fresh vegetables in the suburbs of Shanghai has reached as high as 3.056 million tons. So far, the levels of nitrate and nitrite in fresh vegetables in Shanghai and their possible dietary risks are not clearly known. In this study, we try to test the contents of nitrate and nitrite in fresh vegetables that were randomly collected from five farmers’ markets in Shanghai to study the effects of different fertilization on the accumulation of nitrate and nitrite in the vegetables through a field experiment, and then to probe the variations of nitrate and nitrite contents in the fresh vegetables caused by different patterns of postharvest processing and storage through laboratory tests. We aim to recognize the possible risks of nitrate and nitrite exposures to the human body through daily vegetable intakes in Shanghai and to find methods to reduce nitrate and nitrite in vegetables.

## 2. Materials and Methods

### 2.1. Collection of Vegetable Samples

Fresh vegetable samples, including leafy, root-tuber, and fruit vegetables, 264 in number and 3 in variety, were randomly collected at five farmer’s markets distributed in different areas in Shanghai. Each vegetable sample, about 1 kg in weight, was washed with deionized water, then removed non-edible parts and chemical analyses.

### 2.2. Field Experiment

To study the possible influences of different fertilizations on the accumulation of nitrate in leafy vegetables, a field experiment of growing water spinach (*Ipomoea aquatica Forssk*.) was conducted in the western suburb of Shanghai, Southeast China. The background values of the experimental soil were as follows: Organic matter content, 34.61 g/kg; total nitrogen (N), 2.24 g/kg; alkali-hydrolyzable N, 53.77 mg/kg; total phosphorus (P), 1.45 g/kg; available P, 98.92 mg/kg; total potassium (K), 10.82 g/kg; available K, 321.86 mg/kg.

The experiment included six fertilizer treatments: Synthetic chemical fertilizer in a lower amount (CF1); synthetic chemical fertilizer in a moderate amount (CF2); synthetic chemical fertilizer in a higher amount (CF3); rapeseed cake manure in a lower amount (CM1); rapeseed cake manure in a higher amount (CM2); non-fertilizer control (CK) (see Table 1). The rapeseed cake manure was fermented before its application to the experimental field.

N, P_2_O_5,_ and K_2_O in the synthetic chemical fertilizer applied in the experiment are all 15% in content. Each treatment included basal and topdressing fertilization, accounting for 30% and 70% of the number of treatment fertilizers, respectively. The water spinach was sowed one week after the application of basal fertilizers. The crop was top-dressed half a month after sowing. An experimental plot was 10 m × 10 m in area, and the interval between the two plots was 30 cm. Each treatment was in triplicate. The experimental field was often sprayed to maintain soil moisture. The leaves of water spinach were sampled each week after sowing, carried to the laboratory immediately, and washed with deionized water for chemical analyses.

### 2.3. Laboratory Tests of the Impact of Processing and Storage Conditions

A laboratory test was conducted to study the variation of nitrate and nitrite content in fresh celery (*Apium graveolens* L.) in the whole, fresh-cut, and homogenized forms under the ambient (30 °C), refrigerating (4 °C) and freezing (−20 °C) temperatures during the four-day storage. Another laboratory test was conducted to study the variation of nitrate and nitrite content in freshwater spinach (*Ipomoea aquatica Forssk*.) and green amaranth (*Amaranthus tricolor* L.) samples in whole or homogenized forms under the ambient (30 °C) temperature during the seven-day storage. Each treatment of the tests was in triplicate.

### 2.4. Chemical Analysis of Nitrate and Nitrite in Vegetables

Fresh vegetables were ground and then extracted with deionized water. Nitrate content in vegetables was determined by the salicylic acid-colorimetric method [17], and Nitrite content by the sulfanilamide-N-(1naphthyl) ethylenediamine hydrochloride method [18].

Contents of nitrate and nitrite in the same samples randomly selected were repeatedly measured sixteen times. The relative standard deviations (RSD) for the measurements of nitrate and nitrite were 8% and 5%, respectively, and the recovery rates were 93.8–105.1% and 95.3–108.5%, respectively. In this study, the unit of nitrate content in the samples, mg/kg FW, means NO_3_^−^ mg per kg fresh weight (FW); that of nitrite, mg/kg FW, means NaNO_2_ mg per kg FW.

### 2.5. Standards for Nitrate and Nitrite Contents in Vegetables

According to Chinese national standards [19,20], the nitrate limits for edible permission in leaf, root-tuber, and fruit vegetables are 3000 mg/kg FW, 1200 mg/kg FW, and 600 mg/kg FW, respectively, and the nitrite limits for all are 4 mg/kg FW.

According to a previous study, however, nitrate contamination in vegetables can be classified into four levels (Table 2) [21]. Zhong [22] and Hord [23] also proposed some other levels of classification methods. Compared with the classification criteria, these in Table 2 are still reasonable.

### 2.6. Statistical Analysis

The experimental data were treated with Microsoft Office Excel 2010. One-way analysis of variance was performed by SPSS 10.0 software, and the statistical difference was checked by the Duncan test, with the significance level set to *p* < 0.05. The Origin 9.1 software was used for drawing the figures.

## 3. Results and Discussion

### 3.1. Contents of Nitrate and Nitrite in the Fresh Vegetables in Shanghai

Our investigation showed that nitrate was commonly highly accumulated in fresh vegetables in Shanghai, of which 12.9% were moderately contaminated by nitrate, 13.6% were heavily contaminated, and 25.0% were critically or more contaminated, according to Shen (Table 3) [21]. The nitrate content in the three groups of vegetables is in a descending order of leafy vegetables > root-tuber vegetables > fruit vegetables (see Table 4). Leafy vegetables are more highly enriched in nitrate. The leafy vegetables have many mesophyll cells, including many vacuoles to store nitrate solution. In this study, about 22.6% of the leafy vegetables were higher in nitrate content than 3000 mg/kg FW, the limit for edible permission according to China’s national standards [19].

Nitrate concentration in crops highly varies in different varieties. The leafy vegetables, such as water spinach (*Ipomoea aquatica Forssk*.), green amaranth (*Amaranthus tricolor* L.), red amaranth (*Amaranthus tricolor* L.), celery (*Apium graveolens* L.), spinach (*Spinacia oleracea* L.) and brassica chinensis (*Brassica chinensis* L.), are even more highly enriched in nitrate, in which, the maximum values are 3835.2, 5590.6, 5021.3, 3769.5, 4769.4 and 5511.4 mg/kg FW, respectively. The proportion of nitrate content exceeding 3000 mg/kg FW is 7%, 45%, 73%, 12%, 25%, and 40%, respectively [24]. Alexander [25] also reported a high nitrate concentration in spinach, cabbage, and celery, with the highest value of 4800 mg/kg FW.

However, even the same varieties of leaf vegetables are often significantly different in nitrate concentration. The coefficient of variation (CV) of nitrate content in water spinach, celery, spinach, brassica chinensis, green amaranth, and red amaranth is 80.4%, 86.5%, 77.3%, 41.9%, 51.1%, and 37.7%, respectively (Table 4). This suggests that the level of nitrate accumulation in vegetables is controlled by their genes and highly affected by external factors, such as soil properties and fertilization.

For comparison, the root-tuber and fruit vegetables were generally much lower in nitrate content. Regardless, 6.1% of the root-tuber vegetables exceeded 1200 mg/kg FW in nitrate content, and 5.0% of the fruit vegetables exceeded 600 mg/kg FW in nitrate, which are the limits of edible permission for root-tuber and fruit vegetables, respectively, according to the China national standards [19].

However, the nitrite content of the fresh vegetables in the study is mostly less than 1 mg/kg FW, and none exceeded the nitrite limit [19].

### 3.2. Daily Exposure to Nitrate from Fresh Vegetables

In 2002, the World Health Organization and United Nations of the Food and Agriculture Organization (WHO/FAO) recommended that the allowable daily intake (ADI) of nitrate is 3.7 mg/kg Body Weight (BW). The proper daily nitrate intake should be 222 mg per person if the human body is 60 kg in weight on average.

In this study, nitrate content in the leafy, root-tuber, and fruit vegetables is 1770.41, 536.14, and 476.27 mg/kg FW, respectively. According to the Shanghai residents’ dietary structure, the daily intake of leafy, root-tuber, and fruit vegetables is 151.1 g, 158.8 g, and 108.1 g on average, respectively [26]. The daily nitrate intake per person through eating vegetables in Shanghai attains 404.13 mg, 182% ADI (Table 5), exceeding the WHO/FAO allowable limit.

The WHO/FAO recommends a minimum intake of 400 g of fruit and vegetables per day to prevent chronic diseases [27]. However, human exposure to nitrate is mainly through consuming vegetables, making up approximately 70–90% [28]. Exposure to nitrate via vegetables only exceeded the ADI among Beijing residents and a population in North China [29,30].

The estimated exposure to nitrate for Shanghai residents is higher than in western countries and Korea but is lower than in North China and almost equal to Beijing (Table 6) [29,30,31,32,33,34]. This may be due to Shanghai adults’ higher consumption of leafy vegetables and the relatively high nitrate concentrations in these vegetables. Such estimations, however, may not be completely reliable, as there are large variations in nitrate concentrations among different vegetables, even of the same type. In addition, the exposure data from different studies may not be directly comparable due to differences in study design, calculation methods, and analytical techniques.

### 3.3. Effects of the Different Fertilizations on the Contents of Nitrate and Nitrite in the Vegetables

The field experiment indicated that the content of nitrate in the water spinach for the treatments of CF1, CF2, and CF3 increases sharply, reaching the maximum values one week after top-dressing, and then decreases rapidly (Figure 1). The maxima of nitrate in the vegetables of CF2 and CF3 were nearly 600 mg/kg FW, which still meets the requirements for edible permission [19]. The nitrite in the vegetables of CF1, CF2, and CF3 was higher than 5 mg/kg FW, which exceeded the limit for edible permission [19].

The contents of nitrate in the vegetables for CF1, CF2, and CF3 were increased by applying chemical fertilizers. Moreover, the maxima of nitrate in the vegetables for the treatments were significantly positively correlated with the amounts of fertilizer application (n = 3; *p* < 0.05). This fully shows that the high nitrate content in the vegetables is attributed mainly to the excessive application of chemical fertilizers. After two weeks of topdressing, the nitrate content in the vegetables was sharply reduced and maintained a lower level, within a range of 100–200 mg/kg FW.

For comparison, the maximum values of nitrate in the vegetables of CM1 and CM2 were only 219.5 mg/kg FW and 311.8 mg/kg FW, respectively, which were significantly lower than those of CF1, CF2 and CF3 (n = 3; *p* < 0.05). Moreover, the time to attain the maxima of nitrate content of CM1 and CM2 was approximately one week later than that of CF1, CF2, and CF3 (Figure 1).

Shortly after applying chemical fertilizers, mineral nitrogen and other nutrients are rapidly released into the soil, and nitrate is quickly taken up and temporarily stored in crop vacuoles, thus making it extremely high in vegetables [35,36]. Afterward, nitrate is mostly transformed into ammonium nitrogen for the synthesis of proteins and is highly reduced in content. For food safety, therefore, the vegetables can never be harvested within one week after applying chemical fertilizers.

However, the release of nutrients from organic manure is slow and persistent, resulting in a low level of rapidly available nitrogen in the soil [37]. This explains the lower nitrate content in the CM1 and CM2. Also, for this reason, nitrate is usually less accumulated in the edible part of crops under the organic farming system [38].

### 3.4. Difference in Nitrate Content in the Different Tissues of Vegetables

The content of nitrate in the old leaves of Shanghai pakchoi cabbage (*Brassica chinensis* L.) and Chinese cabbage (*Brassica pekinensis Rupr.*) is 1.7–10.7 times higher than in the young leaves (Table 7). This may be attributed to the difference in metabolic activities between the old and new leaves. Generally, nitrate reductase activity in the old leaves is much lower than in the new, thus causing nitrate accumulation in the old leaves. On the contrary, the strong activity of enzymes in the young leaves makes nitrate quickly metabolized into proteins and less accumulated [39].

Nitrate content in the different parts of vegetable leaves was also highly different. Nitrate in the petioles of Shanghai pakchoi cabbage leaves was 1.1–8.3 times higher than in the blades, and nitrite in the petioles, on the contrary, was significantly lower than in the blades (Table 7).

The nitrate reductase activity in the blades of leaves is 16 times higher than in the petioles [40], which explains the lower nitrate content and higher nitrite in the blades of pakchoi leaves. Nitrate concentration in the petioles is higher than in the other parts of the vegetables. It shows the fastest increase after topdressing nitrogenous fertilizer, while the leaf blades show a slower increase [40]. As the petioles of spinach accumulate more nitrate than the leaf laminae, trimming petioles to increase the lamina/petiole ratio at harvest is necessary to reduce the overall nitrate content of the product [41].

### 3.5. Variation of Nitrate and Nitrite Contents in Fresh Vegetables during Storage

The content of nitrate in vegetables can also be affected by storage and postharvest processing, such as washing, fresh-cutting, and juicing [42]. A laboratory test was performed to study the changes in nitrate and nitrite in the celery during storage. Under an ambient temperature of 30 °C, nitrate content in the celery in the homogenized form was reduced more rapidly than in the whole, and fresh-cut forms and a concomitant abrupt increase in nitrite content in the homogenized celery was observed, which attains as high as 344.47 mg/kg FW after storage for 24 h (Figure 2(a1,a2)). During the first four hours of storage, the content of nitrite in the homogenized celery, possibly transformed from nitrate, has exceeded the limit for edible permission (4 mg/kg FW). For comparison, the content of nitrite in the celery in the whole and fresh-cut forms remains stable and maintains less than 4 mg/kg FW during the four days’ storage at 30 °C.

Stored at the refrigerated temperature of 4 °C, the content of nitrate in the celery in the homogenized form was also sharply reduced, and nitrite synchronously increased to more than 200 mg/kg FW rapidly (Figure 2(b1,b2)). For comparison, nitrate content in the whole and fresh-cut celery at the refrigerated temperature only slightly declined, and nitrite content remained less than 1 mg/kg FW (Figure 2(b1,b2)). Stored at the freezing temperature of −20 °C, the content of nitrate in the celery in the homogenized form was still reduced to some extent during the four days’ storage, and that in the whole and fresh-cut celery is only slightly reduced (Figure 2(c1)). Nitrite content in the celery in the three forms at −20 °C seems unchanged and maintains less than 1 mg/kg FW (Figure 2(c2)).

Another laboratory test was conducted on the storage of water spinach and green amaranth under the ambient temperature of 30 °C. Generally, the content of nitrate in the two vegetables in the homogenized form reduces rapidly, and that of nitrite rises synchronously (Figure 3). Nitrite content in the homogenized green amaranth rises to more than 900 mg/kg FW on the second day of storage. The nitrite content in the water spinach and green amaranth in the whole form was stable on the first two days of the storage but also increased highly after the third day when the vegetables decayed.

The nitrate content in the fresh green amaranth is 3.85 times that of the freshwater spinach. Correspondingly, nitrite content in the green amaranth is significantly higher than in the water spinach after three days of storage (n = 3, *p* < 0.05). A previous study also reported that the enhancement of nitrite content in vegetables during storage is determined by nitrate content [43]. This further suggests that nitrite accumulation in the vegetables during storage is derived from the biological conversion of nitrate.

Changes in nitrate and nitrite in fresh vegetables during storage are highly dependent on temperature because of their great effects on the activities of nitrate and nitrite reductases [25]. As denitrification is very active in ambient environments, it sharply raises nitrite content in vegetables. Chung [44] reported a sharp postharvest decrease in nitrate concentration in fresh spinach, Chinese spinach, and Chinese cabbage after three days at 22 °C, succeeded by an analogous increase in nitrite content. Our studies also show the sharp decrease of nitrate and concomitant increase of nitrite in the homogenized vegetables at ambient temperature during storage (Figure 2 and Figure 3).

Cold storage appears prohibitive to the changes in nitrate and nitrite contents in fresh vegetables. Nitrate content in rocket (Eruca sativa Mill.) is not changed during dark storage for ten days at 4 °C or 15 °C [45]. Especially under freezing conditions, the activities of microorganisms and enzymes are very weak, causing little changes in nitrate and nitrite contents in fresh vegetables. Our work (Figure 2(a1,a2) and Figure 3) is highly consistent with these previous studies.

The vegetables that are mechanically damaged become more vulnerable to the invasion of microorganisms. When the vegetables are processed by the homogenized method, nitrates are released from vacuoles and then converted into nitrites through microbial denitrification [46]. Normally, storing vegetables under refrigeration is relatively safe. However, there is also a risk of increasing nitrite content during the storage of homogenized vegetables under freezing conditions [44]. Our study also indicates that the reduction of nitrate and accumulation of nitrite in the vegetables are greatly affected by the degree of mechanical damages (Figure 2 and Figure 3) and that even occur in the homogenized vegetables at freezing temperature (Figure 2(c1,c2)).

Dietary intakes of a high amount of nitrite through vegetable-based food often occur due to improper storage, which can be up to 400 mg/kg FW, greatly exceeding the food safety limit [47]. Our tests also indicate the sharp increase of nitrite in the homogenized vegetables stored at ambient or refrigerated temperatures (Figure 2(a1,a2,b1,b2)). Therefore, it is urgently necessary to store vegetables with proper means of rapid cooling and optimally low-temperature conditions as well as undamaged forms to prevent nitrite accumulation.

### 3.6. Prevention of Nitrate Exposure Risk through Vegetable Intake

Leafy vegetables were often highly enriched in nitrate (Table 4), which is also proved by many previous studies (Table 8). To reduce the risk of nitrate exposure to the human body, the daily intake of leafy vegetables should be restricted. We highly recommend a small proportion of leafy vegetables in our daily diet. If a person eats 400 g of vegetables per day according to the WHO recommendation, a ratio of leaf, root-tuber, and fruit vegetables had better be 4%:35%:61% [27].

Nitrate and nitrite in leafy vegetables are maximized one week after applying chemical fertilizers (Figure 1). So, leafy vegetables can never be harvested and consumed shortly after applying chemical fertilizers. Lettuce, butterhead lettuce, iceberg lettuce, spinach, cabbage, potato, and tomato under organic farming are much lower in nitrate values than under conventional farming [22]. Organic farming is thus highly encouraged to prevent hyper-accumulation of nitrate in vegetables. Eating organic vegetables is beneficial for preventing nitrate exposure in our bodies.

## 4. Conclusions

About 25.0% of the fresh vegetables in the markets in Shanghai are in critical or more severe levels of nitrate contamination (>1440 mg/kg FW). The leafy vegetables are more highly enriched in nitrate than the root-tuber and fruit vegetables. About 20.3% of Shanghai leaf vegetables have a nitrate content exceeding the limit for edible permission (>3000 mg/kg FW). Residents’ daily intake of nitrate by eating vegetables in Shanghai has exceeded the WHO/FAO allowable limit. However, nitrite content in the fresh vegetables is within the safe level (<1 mg/kg FW).

The excessive application of chemical fertilizers leads to the hyper-accumulation of nitrate and nitrite in the vegetables, and the maxima of nitrate and nitrite are observed one week after the application of chemical fertilizers, which cannot be picked for dietary use. Applying organic manure can reduce the exposure risks of nitrate and nitrite contamination in the vegetables. The risks of nitrate and nitrite contamination in the vegetables can be reduced by applying. The vegetables’ old leaves and leaf petioles are more highly enriched in nitrate due to their weaker metabolic activities.

Vegetables with high nitrate content have a high risk of nitrite contamination during storage due to the biological conversion of nitrate into nitrite. Such risk is easily triggered by suitable temperature and mechanical damage processing. Therefore, it is necessary to store fresh vegetables with proper modes of rapid cooling and undamaged forms to prevent the risks of nitrite accumulation.

## Figures and Tables

**Figure 1 ijerph-19-14487-f001:**
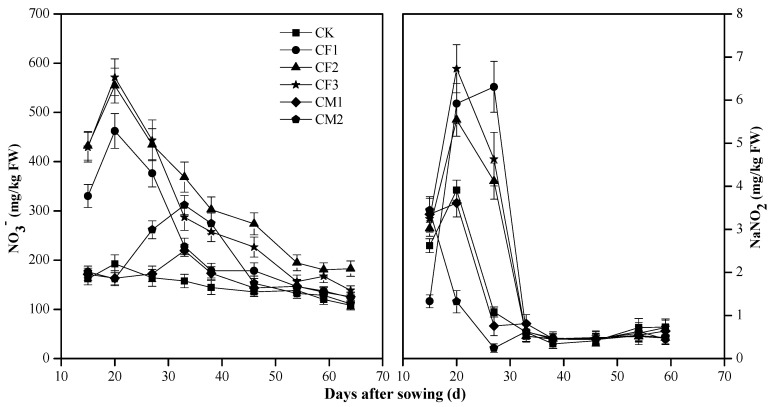
Variations of nitrate (**Left**) and nitrite (**Right**) contents in the water spinach (*Ipomoea aquatica Forssk.*) for the different fertilizer treatments during the growing season of the field experiment. CF1—chemical fertilizer in a low amount; CF2—chemical fertilizer in a moderate amount; CF3—chemical fertilizer in a high amount; CM1—cake manure in a low amount; CM2—cake manure in a high amount; CK—unfertilized control.

**Figure 2 ijerph-19-14487-f002:**
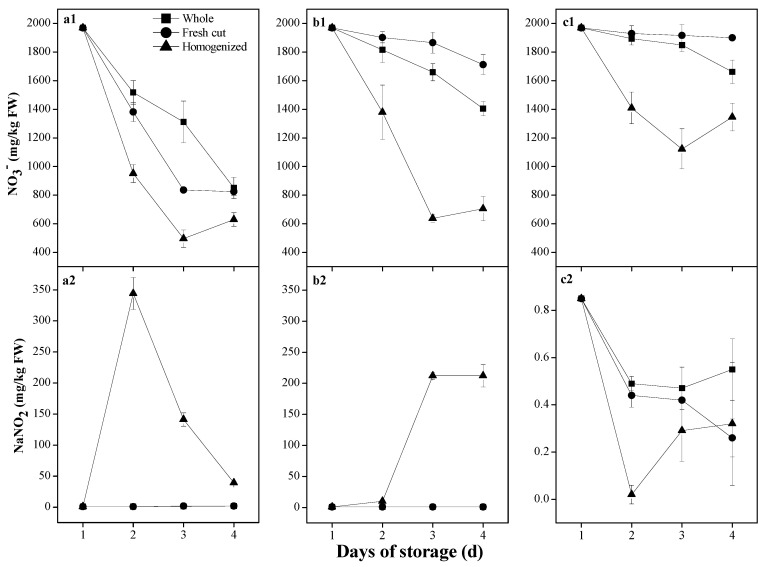
A laboratory test shows that the variations of nitrate and nitrite contents in the fresh celery (*Apium graveolens* L.) in the whole, fresh-cut and homogenized forms stored at temperatures of 30 °C (**a1**,**a2**), 4 °C (**b1**,**b2**) and −20 °C (**c1**,**c2**) for persistent four days.

**Figure 3 ijerph-19-14487-f003:**
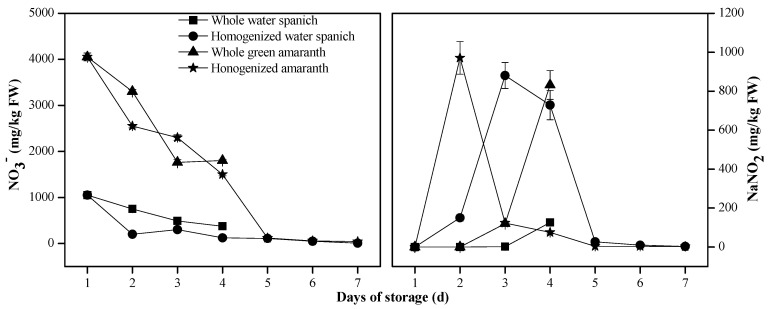
A laboratory test indicates the variation of nitrate and nitrite contents in water spinach (*Ipomoea aquatica Forssk.*) and green amaranth (*Amaranthus tricolor* L.) in the whole and homogenized forms stored at ambient temperature (30 °C) for persistent seven days.

**Table 1 ijerph-19-14487-t001:** Fertilizer applications for the different treatments of the field experiment in the western suburb of Shanghai, Southeast China.

Treatments	Amounts of Fertilizer Application Amounts	Total N	Total P	Total K
	(kg/ha)	(kg/ha)	(kg/ha)	(kg/ha)
CF1	900	135	58.94	112.02
CF2	1800	270	117.88	224.04
CF3	2700	405	176.82	336.06
CM1	2250	135	45.90	22.95
CM2	4500	270	91.80	45.90
CK	0	0	0	0

Note CF1—chemical fertilizer in a low amount; CF2—chemical fertilizer in a moderate amount; CF3—chemical fertilizer in a high amount; CM1—cake manure in a low amount; CM2—cake manure in a high amount; CK—unfertilized control.

**Table 2 ijerph-19-14487-t002:** The classification of nitrate accumulation in fresh vegetables.

Levels	NO_3_^−^ (mg/kg FW)	Contamination Degree	Dietary Limits
I	≤432	Slightly contaminated	Can be eaten raw
II	432–785	Moderately contaminated	Cannot be eaten raw, but can be pickled and cooked
III	785–1440	Heavily contaminated	Cannot be eaten raw or pickled, but can be cooked
IV	1440–3100	Critically contaminated	Cannot be eaten raw, pickled, or cooked, but is not poisonous

**Table 3 ijerph-19-14487-t003:** Levels of nitrate contamination of the three vegetable groups in Shanghai, Southeast China.

Levels	NO_3_^−^ (mg/kg FW)	Total Vegetables (%)	Leaf Vegetables (%)	Root-Tuber Vegetables (%)	Fruit Vegetables (%)
I	≤432	48.5	19.0	56.8	90.0
II	432–785	12.9	13.5	12.5	10.0
III	785–1440	13.6	21.4	10.2	0
IV	1440–3100	14.8	24.6	9.1	0

**Table 4 ijerph-19-14487-t004:** Content of nitrate in the fresh vegetables collected in five farmers’ markets in Shanghai, Southeast China.

Vegetable Category	Name	Sample Number	Nitrate (mg/kg FW)		CV (%)
			Mean	Range	
Leaf vegetables	Lettuce (*var. ramosa Hort.*)	8	255.9	105.7–497	51.9
	Chives stem (*A. tuberosum Rottl. ex Spreng.*)	6	619.5	409.3–876.2	27.1
	Chives (*A. tuberosum Rottl. ex Spreng.*)	8	1055.3	620.8–1908.5	44.5
	Chinese cabbage (*Brassica pekinensis Rupr.*)	9	1135.3	275.3–2176.8	43.6
	Chinese kale (*Brassica alboglabra* L. H. *Bailey*)	6	1142.5	988.1–1507.3	16.7
	Water Spinach (*Ipomoea aquatica* Forssk.)	14	1196.1	139.5–3835.2	80.4
	Green amaranth (*Amaranthus tricolor* L.)	11	2825.5	1055.3–5590.6	51.1
	Celery (*Apium graveolens* L.)	26	1291.4	53.7–3769.5	86.5
	Spinach (*Spinacia oleracea* L.)	8	2098.5	374–4769.4	77.3
	Shanghaipakchoi cabbage (*Brassica chinensis* L.)	15	2891.5	428.8–5511.4	41.9
	Red amaranth (*Amaranthus tricolor* L.)	15	3348.7	280.1–5021.3	37.7
Root-tuber vegetables	Asparagus (*Asparagus Officinalis* L.)	6	686.6	124.8–1712.2	78.5
	Cauliflower (*Brassica oleracea* L. *var. botrytis* L.)	6	1594.0	556.1–2579.8	45.4
	Carrot (*Daucus carota var. sativus Hoffm.*)	8	236.2	101.8–516.6	50
	Radish (*Raphanus sativus*)	8	475.0	202.7–904.1	40
	White turnip (*Raphanus sativus*)	8	1598.1	999.2–2173.0	36.4
	Potato (*Solanum tuberosum*)	10	156.3	40.3–436.9	70
	Taro (*Colocasia esculenta* (L.) *Schott*)	6	161.5	34.5–285.7	60
	Purple potato (*Solanum tuberdsm*)	6	284.9	128.5–594	48.9
	Chinese yam (*Dioscoreae Rhizoma*)	6	504.1	230.9–1245.3	55.9
	Lotus root (*Nelumbo nucifera Gaertn*)	8	162.9	57.8–282.1	40
	Zizanialatifolia (*Zizania aquatica*)	6	181.8	149.5–227	10
Fruit vegetables	Cucumber (*Cucumis sativus* L.)	8	104.7	15.3–190.7	55.6
	Tomato (*Solanum lycopersicum*)	10	105.4	15.35–287.2	64.6
	Chilli (*Capsicum annuum* L.)	6	316.3	211.8–428.5	33.6
	Eggplant (*Solanum melongena* L.)	8	142.2	57.0–331.5	43.1
	Bitter gourd (*Momordica charantia* L.)	8	40.1	15.3–85.7	59.4
	Watermelon *(Citrullus lanatus* (Thunb.) *Matsum. et Nakai)*	6	248.5	157.3–401.2	30
	Muskmelon (*Cucumis melo*)	6	299.6	31.5–692.3	70
	Wax gourd (*Benincasa hispida* (Thunb.) *Cogn.*)	8	502.6	316.9–667.0	20

**Table 5 ijerph-19-14487-t005:** Consumption pattern of vegetables in Shanghai.

Type of Vegetables	Mean Consumption (g day^−1^)	Percentage (%)	Daily Intake of Nitrate (mg)	%ADI
Leaf vegetables	151.1	36.15	267.51	120
Root-tuber vegetables	158.8	37.99	85.14	38
Fruit vegetables	108.1	25.86	51.48	23
Total vegetables	418	100	404.13	182

**Table 6 ijerph-19-14487-t006:** International comparison of average daily exposure to nitrate.

Country/Region	Daily Intake of Nitrate per Person (mg)	%ADI
Beijing, China [29]	330.0	149
France [31]	90.0	41
Korea [34]	102.0	46
New Zealand [32]	31.8	14
United Kingdom [33]	96.0	43
North China [30]	420.0	189
Shanghai, China	404.1	182

**Table 7 ijerph-19-14487-t007:** Content of nitrate and nitrite in the different parts of leaves of Shanghai pakchoi cabbage (*Brassica chinensis* L.) and Chinese cabbage (*Brassica pekinensis Rupr.*).

Name	Parts	Sample Number	Nitrate (mg/kg FW)	Nitrite (mg/kg FW)
Shanghai pakchoi cabbage (*Brassica chinensis* L.)	Old leaves	12	6148.8 ± 92.4	9.22 ± 1.6
	Young leaves	12	1480.5 ± 54.5	1.43 ± 0.4
	Blades of leaves	12	1697.2 ± 35.1	6.61 ± 1.3
	Petioles of leaves	12	4451.6 ± 57.3	2.61 ± 0.3
Chinese cabbage (*Brassica pekinensis Rupr.*)	Old leaves	12	1030.5 ± 34.5	3.17 ± 0.3
	Young leaves	12	267.5 ± 17.1	<1
	Blades of leaves	12	218.1 ± 11.4	<1
	Petioles of leaves	12	812.4 ± 23.1	3.28 ± 0.3

**Table 8 ijerph-19-14487-t008:** Content of nitrate in fresh vegetables previously reported.

Vegetables	Nitrate (mg kg^−1^ FW)		References
	Min–Max	Mean	
**Leaf vegetables**			
Chinese cabbage	429–1610	1300	[48]
	337–3600	2120	[30]
	208–5490	2009	[34]
	232–2236	1243	[49]
	77–1928	933	[28]
	340–2236	1344	[50]
	137–1831	418	[51]
Spinach	340–3650	2090	[50]
	65–8000	2797	[51]
Red amaranth		1180	[30]
	439–3484	2167	[28]
	691–2626	1399	[52]
Lettuce	397–3230	2167	[49]
	21–3986	1074	[53]
	677–2179	1303	[54]
Chives	863–9323	1020	[34]
Celery	446–10,800	3600	[30]
	256–830	565	[49]
	18–3319	1103	[28]
	256–1113	660	[50]
		2110	[55]
	20–4296	1496	[51]
**Root-tuber vegetables**			
Carrot	7–1042	264	[53]
	21–1574	296	[28]
		503	[3]
Potato	2–704	158	[53]
	10–340	168	[28]
		102	[55]
Radish	670–1500	1309	[49]
	766–4570	2108	[34]
Lotus root		120	[48]
**Fruit vegetables**			
Tomato	190–347	238	[48]
	10–259	78	[30]
		392	[23]
		36	[55]
Watermelon		95	[49]
		33	[48]
Wax gourd	358–680	541	[48]
Cucumber	4–245	93	[53]
	30–1236	160	[49]
	22–409	185	[28]
	89–740	335	[50]
Eggplant	67–1000	479	[30]
	29–572	314	[28]
		302	[55]

## Data Availability

The datasets used and/or analyzed during the current study are available from the corresponding author upon reasonable request.

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
