# Peer review of "Nitrate Quantification in Fresh Vegetables in Shanghai: Its Dietary Risks and Preventive Measures"

_ijerph, 2022, doi:10.3390/ijerph192114487_

Round 1

Reviewer 1 Report

1. I have noticed that English language use is not precise in many sentences; I highlighted some in yellow. Grammarly or other proof checkers may help, or have a native English speaker read it again.

2. 2.2 Field experiment----- light and irrigation conditions?

3. 2.3 Laboratory tests ---- please rephrase this sub-title, something like "The impact of processing and storage conditions"?

4. 2.4 Chemical analysis ----  "Nitrate and nitrite analyses"? And what is the extraction method?

5. Section 2.4 ----  which part of the vegetables was sampled for NOx analyses? I would suggest the authors use the ion/salt format for both no3 and no2.

6. Section 2.5 ----  please double-check whether the cited standards are still effective or have been terminated or replaced by new standards.

7. Table 2 ----  NO3 numbers cited were just from one study; a better classification is needed: Zhong, L., Blekkenhorst, L. C., Bondonno, N. P., Sim, M., Woodman, R. J., Croft, K. D., Lewis, J. R., Hodgson, J. M., & Bondonno, C. P. (2022). A food composition database for assessing nitrate intake from plant-based foods. Food chemistry394, 133411. https://doi.org/10.1016/j.foodchem.2022.133411

Norman G Hord, Yaoping Tang, Nathan S Bryan, Food sources of nitrates and nitrites: the physiologic context for potential health benefits, The American Journal of Clinical Nutrition, Volume 90, Issue 1, July 2009, Pages 1–10, https://doi.org/10.3945/ajcn.2008.27131

8. section 3.1 and others ---- plants will accumulate NOx naturally; what are the levels that are defined as "contaminated" or "enriched"?

9. Table 4 ----  please check scientific names. The lettuce, asparagus and cauliflower values are questionable. Therefore, a comparison with existing literature would be beneficial. Asparagus and cauliflower are not leafy vegetables; watermelon is a fruit.

10. Section 3.2 ---- Please double check the WHO ADI, it is not 5 mg/kg bw.

11. "The permissible daily intake of nitrate should be 300 mg per person if defining an average weight of the human body as 60 kg.” ---- please show your calculations for this statement.

12. "the daily intake of the leafy, root-tuber and fruit vegetables is 151.1 mg, 158.8 mg and 108.1 mg on average, respectively", please double check. 

13. To calculate NOx intake from veg consumption, the consumption amount of each vegetable is required. Using each group's mean values is unreliable, given the apparent wide variations and skewness within each group.

14. ‘Shortly after the application of chemical fertilizers, mineral nitrogen and other nutrients are rapidly released into soil, and nitrate is quickly taken up and temporarily stored in crop vacuoles, thus making it extremely high in the vegetables. Afterwards, nitrate is mostly transformed into ammonium nitrogen for the synthesis of proteins and is highly reduced in content. For food safety, therefore, the vegetables can never be harvested within one week after application of chemical fertilizers.’ ---- references please.

Reviewer 2 Report

The paper presents an interesting evaluation of the nitrate and nitrite accumulation in different vegetables from Southeast China.

 The authors are recommended to consider the following aspects:

·         The title form should be reconsidered. A more appropriate state could be: ”Nitrate quantification from……”, ”Determination of nitrate quantities in…”;

·         Overlook the entire material and reformulate the phrases. Some words are overused even though they represent the article's main points.

·         Since it was concluded that natural manure is preferable to the synthesis products, it could be considered to present the necessary preliminary treatments that need to be done before its application in the field.

·         The presence of these compounds in the daily diet is a reality that can not be neglected. In such a situation, the authors could consider the data obtained and corroborate it with the ones presented in the literature. The approach targets developing a plan to reduce the studied substance's regular ingestion. In this regard, different strategies could be considered: total or partial replacement of some products with others; the maximum quantity of merchandise used considering the processing performed;  the parameters of manufacturing; etc. The authors could also consider other criteria to strengthen the proposal.

·         Valuable current references should sustain all advice.

The paper could be considered for publication after major changes. It has to be revised by the authors and resubmitted with suggested modifications specified in the reviewer’s comments.

Round 2

Reviewer 1 Report

Thanks for the responses and the great effort. It is a pleasure to see the manuscript has been well improved. Just a few comments:

1.       Comment 7. Thanks. Could you please add your references that support your statements like “Cannot be eaten raw”? The eating amount might be more critical. For example, rucola has around 5000 mg/kg of nitrate but it is a salad leafy veggie and eaten raw. I don't think there are recommendations about vegetable nitrate, except that the EU has regulations on some vegetables. In contrast, more recent studies have suggested the health benefits of leafy vegetable nitrate. See Zhong (2022), Hord (2009) and J.O. Lundberg, M. Carlstrom, E. Weitzberg. Metabolic effects of dietary nitrate in health and disease. Cell Metabolism, 28 (1) (2018), pp. 9-22, 10.1016/j.cmet.2018.06.007. In fact, beetroot juice is being used as a food supplement in many studies.

2.       Table 4. Some of the words/letters of the Latin names (such as L.) should not be in italic. Please double-check.

3.       Table 4. Water spinach.

4.       Table 4. Asparagus should be a stem/stalk vegetable? Cauliflower is a flower vegetable. Please see J.A.T. Pennington, R.A. Fisher. Classification of fruits and vegetables.Journal of Food Composition and Analysis, 22 (2009), pp. S23-S31, 10.1016/j.jfca.2008.11.012

5.       Comment 10: WHO recommended nitrate (ion) is 3.7 mg/kg bw. Please double check, and amend Comment 11.

6.       Comment 12, you re-grouped some of your vegetables in Table 4, the values would be different?

7.       Comment 13, OK, then I may suggest the authors use 1-2 sentences to discuss this limitation.

Reviewer 2 Report

The authors have considered a few of the recommendations previously made. Some aspects, however, need attention.

·         The wording of the title still needs to be refined. It isn't very clear.

·         The overused words situation remains unchanged. Try to rephrase the sentences or use synonyms.

·         The reference section was very slightly improved.

·         The following suggestion: ” The presence of these compounds in the daily diet is a reality that can not be neglected. In such a situation, the authors could consider the data obtained and corroborate it with the ones presented in the literature. The approach targets developing a plan to reduce the studied substance's regular ingestion. In this regard, different strategies could be considered: total or partial replacement of some products with others; the maximum quantity of merchandise used considering the processing performed;  the parameters of manufacturing; etc. The authors could also consider other criteria to strengthen the proposal”, it is not found in the uploaded material.

  The paper could be considered for publication after major changes. It has to be revised by the authors and resubmitted with suggested modifications specified in reviewer’s comments.
